# From an Image to a Scene: Learning to Imagine the World from a Million 360° Videos

**Matthew Wallingford**[◇]
**Anand Bhattad**[†]   **Aditya Kusupati**[◇]   **Vivek Ramanujan**[◇]   **Matt Deitke**[◇]
**Sham Kakade**[△]   **Aniruddha Kembhavi**[‡◇]   **Roozbeh Mottaghi**[◇]   **Wei-Chiu Ma**[‡◇]   **Ali Farhadi**[◇]
[◇]University of Washington   [†]Toyota Technological Institute at Chicago
[△]Harvard University   [‡]Allen Institute for AI
mcw244@cs.washington.edu

## Abstract

Three-dimensional (3D) understanding of objects and scenes play a key role in humans' ability to interact with the world and has been an active area of research in computer vision, graphics, and robotics. Large scale synthetic and object-centric 3D datasets have shown to be effective in training models that have 3D understanding of objects. However, applying a similar approach to real-world objects and scenes is difficult due to a lack of large-scale data. Videos are a potential source for real-world 3D data, but finding diverse yet corresponding views of the same content has shown to be difficult at scale. Furthermore, standard videos come with fixed viewpoints, determined at the time of capture. This restricts the ability to access scenes from a variety of more diverse and potentially useful perspectives. We argue that large scale 360° videos can address these limitations to provide: *scalable corresponding frames from diverse views*. In this paper, we introduce 360-1M, a 360° video dataset, and a process for efficiently finding corresponding frames from diverse viewpoints at scale. We train our diffusion-based model, ODIN[1], on 360-1M. Empowered by the largest real-world, multi-view dataset to date, ODIN is able to freely generate novel views of real-world scenes. Unlike previous methods, ODIN can move the camera through the environment, enabling the model to infer the geometry and layout of the scene. Additionally, we show improved performance on standard novel view synthesis and 3D reconstruction benchmarks.

## 1   Introduction

Humans have the ability to understand and reason about the 3D geometry of the world, which is key for everyday tasks such as navigation and object manipulation [14, 67, 62, 48]. In machine learning, 3D perception and reasoning has been a long-standing goal for researchers with broad applications in robotics [58, 20, 61], vision [27, 25], and graphics [22, 57]. Fueled by large-scale datasets of synthetic objects [10, 11], recent generative models have shown impressive understanding of 3D objects [24, 44, 31]. While these models' ability to generate synthetic objects is impressive, enabling 3D generative models for real world scenes and objects remains an open challenge.

One intuitive source for scalable data has been video as it implicitly contains rich information about the 3D world. However, learning 3D modeling from video has been elusive despite impressive effort [66, 36, 19, 37, 38]. The key problem has been how to consistently transform video into a form amenable to learning about the 3D world. Existing 3D models learn from multi-view data, a

---

[1]In Norse mythology, Odin uses his ravens, Huginn and Muninn, as his eyes to fly throughout the world and relay what they see.

38th Conference on Neural Information Processing Systems (NeurIPS 2024).

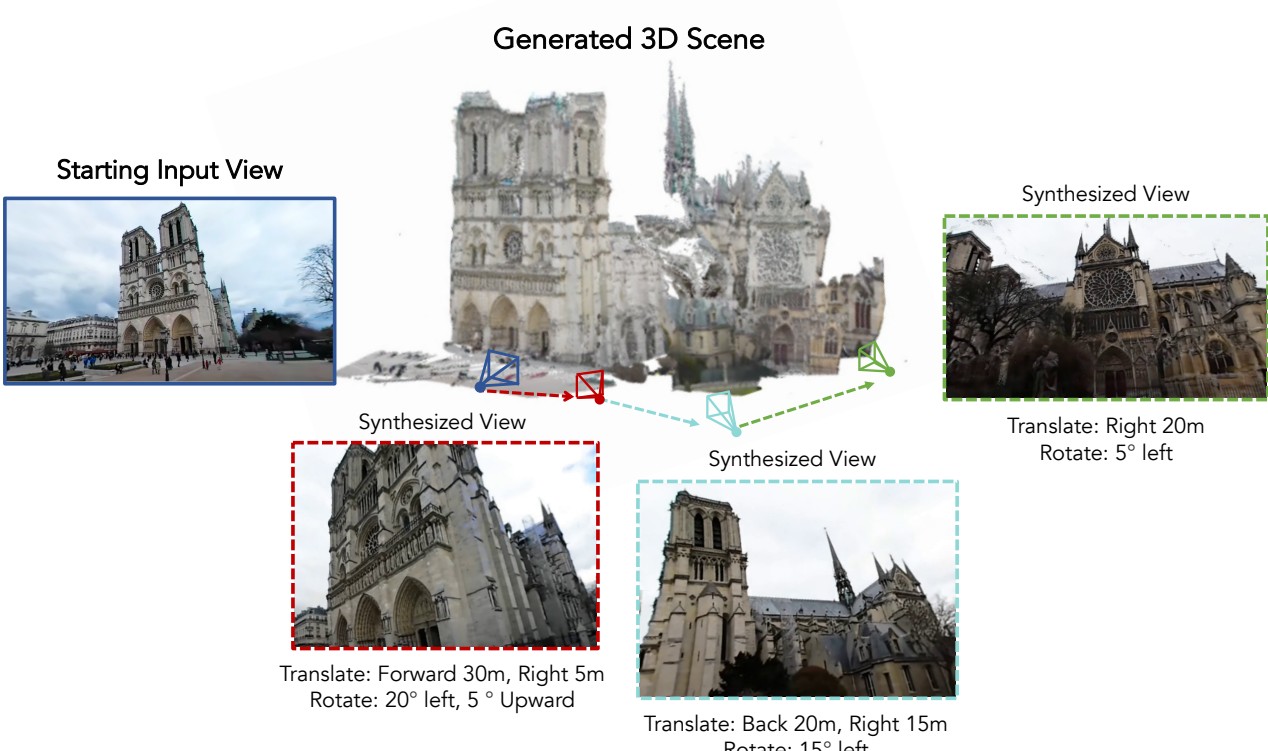

Generated 3D Scene

Starting Input View

Synthesized View

Synthesized View
Translate: Right 20m
Rotate: 5° left

Synthesized View
Translate: Forward 30m, Right 5m
Rotate: 20° left, 5 ° Upward

Synthesized View
Translate: Back 20m, Right 15m
Rotate: 15° left

Figure 1: By learning from the largest real-world, multi-view dataset to date, our model ODIN, can synthesize novel views of rich scenes from a single input image with free camera movement throughout the scene. We can then reconstruct the 3D scene geometry from these geometrically consistent generations.

collection of images of scenes or objects and their respective camera pose. Creating such multi-view datasets from video requires finding sets of corresponding frames that capture similar parts of the scene but from different locations (Figure 2).

This search for corresponding frames in video has proven difficult at scale for a few reasons. First, correspondences are sparsely distributed throughout the video because the trajectory of the camera is fixed at the time of capture. Ideally, the camera operator would focus on a specific object or portion of the scene while moving around it. However, in-the-wild videos are far from this ideal. For example, if a person records themselves walking in the park, it is rare that they consistently focus the camera on the same object such as a bench as they walk towards, past, and away from it. Second, the computational cost of checking whether frames form a correspondence is expensive [42, 40, 5], therefore searching extensively is infeasible. Given these limitations, the largest real-world multi-view datasets to date [34, 65] utilize Amazon Mechanical Turkers to manually record video clips of objects, and are limited to 50 and 238 object categories respectively.

To address these limitations, we collect one million 360° videos from YouTube, introduce a process to efficiently transform 360° video into multi-view data, and train a diffusion-based novel-view synthesis (NVS) model on the dataset. Our model named ODIN, is the first to reasonably synthesize real-world 3D scenes and reconstruct their geometry *conditioned on a single image*. Quantitatively we evaluate our method on standard novel view synthesis benchmarks (DTU and MipNeRF360) and find improved performance compared to existing models without fine-tuning our own. Additionally, we compare ODIN to existing methods for 3D reconstruction on Google Scanned Objects as well

as a held-out set of 360-1M and show significantly improved performance, especially on complex real-world scenes. We will open-source our model and dataset.

## 2   Related Work

**Novel View Synthesis.**    NeRF [26] optimizes a volumetric scene function using sparse 2D images, representing the scene as a continuous 5D function. MipNeRF [3] extends NeRF with a multi-scale representation to enhance detail and reduce aliasing. Plenoctree [63] combines NeRF principles with an octree structure for efficient rendering. DIVeR [55] proposes a deterministic volumetric rendering for NeRF. Gaussian Splatting [18] uses Gaussian functions and splatting techniques for detailed scene representation and rendering. Unlike these methods which rely on densely sampled multi-view images and known camera poses, our approach captures extensive real-world scenes from widely varying camera views. PixelNeRF [64] and DietNeRF [16] extend NeRF to handle sparse input views but only for controlled settings.

Recent works leverage powerful generative (diffusion) models [35] for novel view synthesis of objects [31, 21, 53, 6], and more recently for scenes [6, 39, 8]. ZeroNVS uses a 3D-aware diffusion model with novel camera conditioning to generate 360-degree views from a single image, focusing on depth-scale ambiguity and background diversity with synthetic and real-world datasets. Diffusion with Forward Models [47] integrates a forward model into the diffusion process for unsupervised training on partial observations, solving inverse problems like view synthesis without direct signal supervision. ReconFusion [56] combines NeRFs with diffusion priors to enhance 3D reconstruction from limited views, improving geometry and texture plausibility with real and synthetic multi-view datasets. LucidDreamer [8] and RealmDreamer [45] use a multi-step pipeline involving point cloud guidance and Gaussian splats to generate detailed 3D scenes from text or image prompts but lacks physical realism and has limited control over viewpoint changes. In contrast, our method leverages a large-scale collection of 360-degree YouTube videos to train a diffusion-based model, enabling the synthesis of diverse real-world 3D scenes and reconstruction from a single image, thus accommodating significant camera view changes and a broader range of scenarios.

**Camera Pose Estimation and Structure from Motion.**    Estimating camera pose and structure-from-motion (SfM) have a rich history in computer vision [2, 43, 50]. Camera pose estimation consists of estimating the 6 degrees of freedom of cameras from which images were taken and the camera intrinsics. The process typically involves finding corresponding key-points between multiple images of a scene, and using their apparent motion within the images to infer the 3D geometry of the scene and relative location of each camera. For multi-view datasets and novel view synthesis, works typically use COLMAP [42, 40] or a SLAM variant [46, 28]. We choose to use the recent method Dust3R [52] as it is computationally faster and allows for as few as 2 images whereas most SfM methods require dozens. This enables us to scan much more quickly through videos for frame correspondences and create the large-scale dataset from 360° video.

**Multi-View Datasets.**    Existing multi-view datasets such as MVImageNet [65], CO3D [34], RealEstate10K [68], ACID [23], Epic-Kitchens [9], MipNeRF-360 [4], and Epic-Fields [49] provide valuable multi-view sequences of real-world scenes and objects but are often constrained by the specific environments or objects they capture. MVImageNet is the largest multi-view dataset to date with over 200,000 video clips captured of 238 object categories. Though this effort is impressive, using Mechanical Turkers to manually capture videos of objects is difficult to scale further and limits content diversity. In Figure 10 we show examples of correspondences within MVImageNet which can be compared to correspondences of 360-1M in Figure 8. Large-scale 3D object datasets like Objaverse [10], Objaverse-XL [11], and infinigen [32] focus on detailed 3D object assets to generate synthetic objects and scenes. Autonomous driving and 3D reconstruction datasets such as Kitti [15], DTU [1], ShapeNet [7], and Google Scanned Objects [13] offer multi-view data for specific tasks like driving scenarios, 3D modeling, and object classification.

In contrast, our dataset leverages a large-scale collection of 360-degree YouTube videos, providing a vastly more diverse and extensive source of real-world data. Our dataset accommodates significant camera view changes and broader real-world applications, going beyond the constraints of controlled multi-view datasets and specific domain focuses.

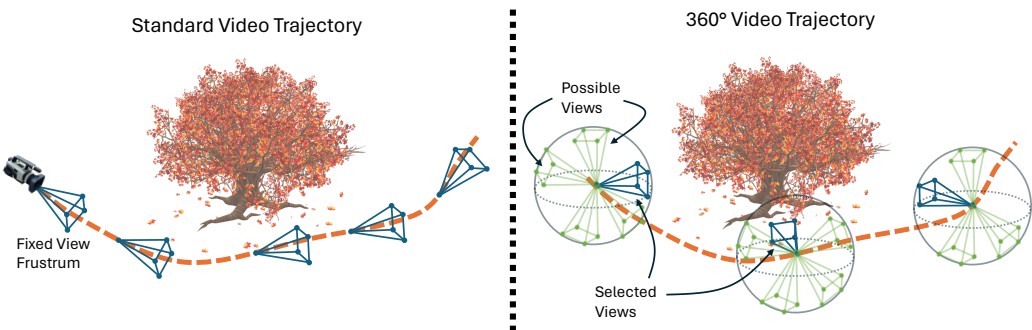

Figure 2: Left: An illustrative trajectory of standard video with the view point fixed at the time of capture. The fixed view point makes finding corresponding frames challenging. Right: The trajectory of a 360° video through the scene. The controllable camera enables alignment of views at different frames of the video.

# 3   Multi-View Data from 360° Video

There are two key elements missing from current multi-view datasets: *scale* and *real-world* data. Various datasets and works have managed [34, 65, 10, 11] to make progress along these dimensions individually, however, no current datasets afford both aspects.

The key challenge in collecting large-scale, multi-view datasets derives from the difficulty of finding high-quality frame correspondences, and estimating their relative poses. Existing structure-from-motion algorithms, such as COLMAP [41] and HLOC [40], are slow and require many images of the same scene. In this section we detail our process for efficiently transforming 360° video into high-quality multi-view data.

## 3.1   Scalable Correspondence Search

There are two properties of corresponding video frames that are necessary for training novel view synthesis models: *sufficiently differing viewpoints* and *overlapping content*. In manually collected novel view synthesis (NVS) datasets this is accomplished by taking a video while circling the object. Finding frames that fit these criteria from in-the-wild video is much more difficult.

A major reason is that high-quality correspondences are sparsely distributed in standard videos. For example, someone taking a video while walking down the street often keeps their camera view facing their direction of travel. So while they may capture a parked car on the side of the road while walking towards it, they likely will not pan their camera to capture it from many angles while walking past or away from it. Therefore, it is difficult to obtain paired images of the scenes or objects from distant locations and diverse views at scale. One solution to this problem is to leverage 360° videos. The 360° nature allows the views of frames to be rotated such that they contain overlapping content. Therefore given two frames that are close enough in spatial location, in theory we can align the views to look at the various regions of the scene to form multiple view correspondences.

Now we describe how we operationalize this approach. We begin by sub-sampling frames of the 360° video at $r = 1$ frame-per-second. We find empirically this to be a sufficiently fast frame rate given the movement speed of the camera. The computation of the correspondence search scales with $r^2$, therefore we judicially select the frame rate. Next we perform pairwise comparison between frames within a frame window of length, $L = 20$. We map the 360 panoramic frames using an equirectangular projection $E(I, \theta, \phi)$ where $\theta$ is the pitch, $\phi$ is the yaw, and $I$ is the image. We map the panoramic image to four different views $E(I, j * \pi/2, 0)$ for $j \in \{1, \ldots, 4\}$. Thus a panoramic frame, $F_t$ at time $t$ produces four frames $\{F_{t,0}, F_{t,\pi/2}, \ldots, F_{t,3*\pi/2}\}$. We then pass all pairs within the time window to the Dust3r model [52] which outputs relative pose estimate, $P$ and confidence map, $C$. We take the mean confidence over the spatial components of the confidence map with height $h$ and width $w$, $\mu_c = \frac{1}{hw} \sum_{c \in C} c$, and filter out frames below threshold, $\tau = 4$. A higher mean confidence means that the frames must be overlapping as the model can accurately estimate the pose.

Once the correspondences have been found we refine the relative pose between them by performing gradient descent on the pitch and yaw of both equirectangular projections with respect to $\mu_c$. Intuitively, we can think of this as rotating the cameras to maximize the overlap (Figure 2). After all correspondences have been found, we discard pairs with relative translation less than .25 m because they provide minimal information for training the model.

## 3.2 Correspondence Propagation

Computing relative pose between frames, especially for video, has been computationally prohibitive and a major bottle-neck for large-scale multi-view datasets [9, 34, 65]. An exhaustive search between all frames of a video would incur a cost of $s^2r^2$ where $s$ is the number of seconds, and $r$ is the frame rate. A common approach is to limit the search with a window of size $L$ to reduce the cost to $L^2$, however this limits the pairs to short-range correspondences.

We propose a hybrid approach that enables finding long-range correspondences with limited additional compute. After the initial frames have been found as detailed in section 3.1, we create a graph in which the nodes are frames and an edge exists if two frames have correspondence. We then perform the same procedure outlined in section 3.1 for all connected frames in each sub-graph. Intuitively, if two frames share a corresponding third frame (connected in the graph)then the two are also likely to share a correspondence.

This approach allows us to maintain a small search window, while still finding long-range correspondences. We show examples of such long-range correspondences in Figures 8 and 9.

## 3.3 Resolving Scale Ambiguity

Dust3R and other structure-from-motion methods [29, 41, 33] output relative camera poses in dimensionless quantities, therefore we need to calibrate them to a universal scale. We do so by fusing the depth map estimates, $\hat{D}$, from an off-the-shelf depth estimator [60], with the point map, $X \in \mathbb{R}^{h \times w \times 3}$, predicted by Dust3R. A pointmap is a correspondence between each pixel $(i, j)$ and the point in 3D where the ray from pixel $(i, j)$ intersects the scene. We anchor the dimensionless pointmap to the depth math $D$ by optimizing for a scale factor, $\sigma$, in the following equation:

$$\arg\min_{\sigma} \sum_{i=1}^{h} \sum_{j=1}^{w} |C_{ij}(\sigma z_{ij} - D_{ij})|, \tag{1}$$

where $z_{ij}$ is the depth component of $X_{ij}$ and $C_{ij}$ is the confidence map output by Dust3R. $C_{ij}$ is close to 0 for points which the model has high uncertainty and acts as a filter for points with poor estimates. We choose L1 distance to limit the effect of outliers. Once we recover this scale factor, we multiply the translation of the estimated camera pose $(R, t)$ by $\sigma$ to obtain the metric pose estimate.

## 4 Dataset Collection and Statistics

To leverage the proposed scalable correspondence search (Section 3) for generating a large-scale multi-view dataset, we collect the largest 360° video dataset to date, 360-1M, consisting of over 1 million 360° videos. In this section, we describe the collection process and statistics for the dataset.

### 4.1 Collecting 360° Video

We collect all meta-data from YouTube in order to filter 360° videos. The meta-data provides information on duration, view count, format, and subject category among other fields. We filter for the equirectangular format which indicates 360° video and results in 1,076,592 total videos. We then download the videos in the equirectangular format at the best quality available. We will release the meta-data for the 360° videos alongside the dataset.

We filter the downloaded videos for empty, and duplicate videos. We remove duplicated videos with a deduplication model [17] run on the thumbnails of the videos. This does not guarantee the contents of the video are unique, however running over all frames is computationally infeasible.

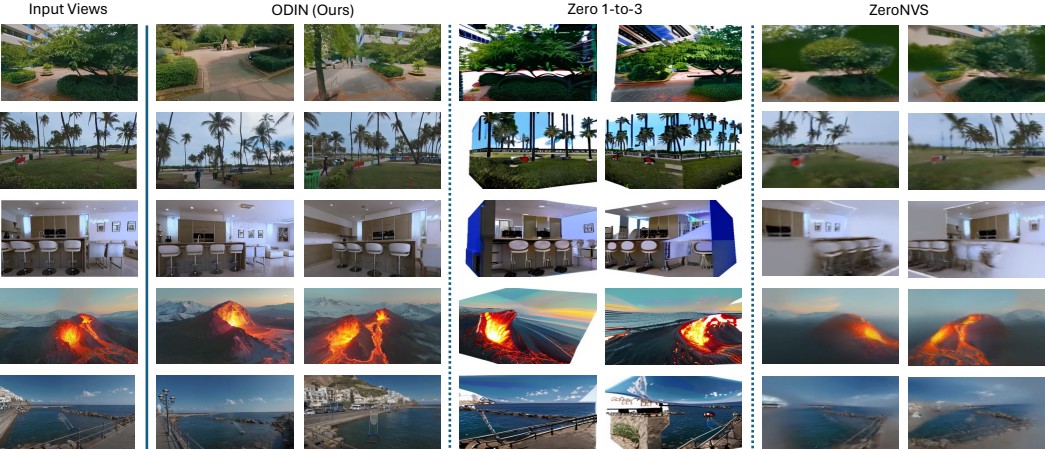

| Input Views | ODIN (Ours) | Zero 1-to-3 | ZeroNVS |

Figure 3: Qualitative comparison of novel view synthesis on real-world scenes. The left and right images are conditioned on camera views from the left and right respectively. In the middle scene of the kitchen, ODIN accurately models the geometry of the table counter and chairs as well as unseen parts of the scene such as the living room.

## 4.2 Dataset Statistics

360-1M consists of 80,567,325 unique frames extracted from 1,076,592 videos with an average of 74.83 unique frames per video. The average video length is 6.3 minutes and is distributed in a long-tail fashion (Figure 5). When searching for correspondences, we sample the videos at 1 FPS. The videos are distributed evenly across 15 subject categories, with the most popular category being Travel and Events (149,534 videos) and the least popular being Pets and Animals (8802 videos) (Figure 6). We find 363,417,730 total frame correspondences along with their relative camera poses.

## 5 Method

Our final goal is to generate images along a viewpoint trajectory conditioned on a single image of a scene – a task known as novel view synthesis (NVS). Note that our task differs from tradition novel view synthesis work such as [26] which aim to generate novel views after training on many images of a single scene. Similar to prior works [39, 24], we leverage a diffusion-based model. This class of models have shown impressive capabilities in learning priors from large-scale data. An alternative is a NeRF based approach which is mainly effective in small-scale settings.

### 5.1 Viewpoint Conditioned Diffusion

Given a single image, $x \in \mathbb{R}^{h \times w \times 3}$, of a scene, our objective is to generate a sequence of images, $\hat{x}_i$ from different viewpoints, $(R, t)$ where $R$ is the relative rotation and $t$ is the translation between views. Following [24], we use a latent diffusion architecture which consists of an encoder $\mathcal{E}$, a denoiser U-net $f_\theta$, and decoder $\mathcal{D}$. The standard diffusion training objective is:

$$\min_\theta \mathbb{E}_{z \sim \mathcal{E}(x), t, \epsilon \sim \mathcal{N}(0,1)} \left\| (\epsilon - f_\theta(z_t, t, f(x, R, t))) \right\|_2^2$$

Our modeling objective differs from previous work in that we condition on both rotation, $R$, and translation $t$. The long-range correspondences in our training data afford much freer camera movement throughout the scene compared to previous works. Due to the limitations of previous training data, other methods can only rotate about a center point of the object or scene.

### 5.2 Motion Masking

Learning how to perform novel view synthesis from videos poses a challenge as it assumes the scene itself does not vary with time when generating images from novel viewpoints. Previous approaches

Figure 4: Examples of generated 3D scenes using ODIN. The blue dot indicates the location of the input image and the red lines indicate the trajectory of the camera which generated the images. ODIN is capable of long-range generation of geometrically consistent images. In the bottom scene, we see the model accurately infers the geometry of the unseen cathedral ceiling and the long hallway.

have addressed this challenge by training solely on videos of static scenes such as only indoor houses [68] or manually filtering videos [34, 65]. However, such approaches limit the diversity and scale of the data. Therefore, to learn from in-the-wild videos, we propose *motion masking*, an approach for handling dynamic objects.

Motion masking consists of predicting a dense mask of values between 0 and 1, which we apply to the output by the U-net $f_\theta$ through elementwise multiplication. This soft mask allows the model to filter out portions of the scene which may be difficult to predict due to object movement. To produce the motion mask we add an additional channel to the U-Net denoiser, which outputs a dense mask with values which we clamp between 0 and 1. During training, this mask filters dynamic elements from the loss function.

Formally, let $M \in \mathbb{R}^{h \times w}$ denote the dense mask generated by the decoder. The modified loss function, incorporating temporal masking, is given by:

$$\mathcal{L} = \|(\epsilon - \epsilon_\theta(z_t, t, f_\theta(x, R, t))) \cdot M\|_2^2 \tag{2}$$

However, directly optimizing this loss leads to a degenerate solution where all elements of the scene are filtered from the loss. To address this, we introduce an auxiliary loss term that incentivizes the mask to be non-zero:

$$\mathcal{L}_{\text{auxiliary}} = -\lambda \sum_{i,j} M_{ij} \tag{3}$$

Incorporating motion masking and the auxiliary loss enables the model to focus on static elements in dynamic scenes while training for novel view synthesis.

## 5.3 3D Reconstruction

Our model is trained to output a single image given an input image and target view, a popular approach which provides flexibility in the type of data that can be trained on, while still allowing for 3D reconstruction and multi-view generation. This flexibility is particularly crucial for training from video data, where obtaining a full collection of frames for a given scene from in-the-wild videos may not be possible.

Naively, the image to image paradigm has the drawback that generating multiple views does not guarantee consistency across views. To address this, we follow the approach of previous works [39, 31, 24] which employ various techniques to induce consistency across multiple generations. We adopt a trajectory-based sampling approach similar to [39] where images are sampled along a smooth trajectory, though in our case we are not restricted to simple rotations. While sampling, subsequent generations are conditioned on the previous generation, $\epsilon_{i,t} = f_\theta(x_i, R, t)$. Once multiple views are generated we reconstruct the scene using Dust3r [52].

# 6 Experiments

In this section we benchmark our model, ODIN, against existing methods for novel view synthesis and 3D reconstruction. We improve performance on standard benchmarks which consist of relatively simple scenes with minimal camera translation, all without fine-tuning on the target task. Qualitatively we find that ODIN has new capabilities in generating real-world scenes from long-range novel views.

## 6.1 Experimental Setup

We evaluate our model on the standard novel view synthesis (NVS) benchmarks, DTU [1], and Mip-NeRF 360 [4]. DTU consists of table-top items and Mip-NeRF 360 consists of scenes with views rotated 360° around a point. We report the standard NVS metrics, PSNR, LPIPS, and SSIM. As noted by previous literature, PSNR and SSIM are not well correlated with human evaluation so we primarily focus on LPIPS and qualitative comparison. Furthermore, to showcase the novel capabilities of our model, we evaluate our method on a held-out set of 360-1M constructed from one-thousand 360° videos.

For 3D reconstruction we compare with Zero1-to-3 [24], MCC, SJC-I, and Point-E on Google Scanned Objects (GSO) and ZeroNVS on our held-out set of 360-1M. For 360-1M we derive the pseudo-ground truth from a Dust3R model which is trained on all ground truth views of the scene given by the video. We report Chamfer-Distance for 360-1M in addition to volumetric IoU for GSO. The 3D reconstructions for our model are created by generating images along trajectories then using Dust3r to reconstruct the scene.

Table 1: Comparison with other novel view synthesis models on the DTU benchmark which consists of single objects placed on table tops.

| NVS | LPIPS ↓ | PSNR ↑ | SSIM ↑ |
|---|---|---|---|
| PixelNeRF [64] | 0.535 | 15.55 | 0.537 |
| SinNeRF [59] | 0.525 | 16.52 | **0.560** |
| DietNeRF [16] | 0.487 | 14.24 | 0.481 |
| NeRDi [12] | 0.421 | 14.47 | 0.465 |
| ZeroNVS [39] | 0.380 | 13.55 | 0.469 |
| ODIN (Ours) | **0.378** | **16.67** | 0.525 |

Table 2: Comparison of various novel view synthesis models on the MipNeRF 360 benchmark [39, 4]. As noted by previous work [39], PSNR and SSIM are unreliable metrics for novel view synthesis so we focus on LPIPS.

| NVS | LPIPS ↓ | PSNR ↑ | SSIM ↑ |
|---|---|---|---|
| PixelNeRF [64] | 0.718 | 16.50 | **0.556** |
| Zero-1-to-3 [24] | 0.667 | 11.70 | 0.196 |
| ZeroNVS [39] | 0.625 | 13.20 | 0.240 |
| ODIN (Ours) | **0.587** | **16.84** | 0.537 |

The models we benchmark against are trained on a variety of 2D and multi-view data sources. The diffusion-based methods, Zero1-to-3 [24] and ZeroNVS [39] start from a StableDiffusion pretrained model. Zero1-to-3 [24] fine-tunes on Objaverse [10], while ZeroNVS [39] fine-tunes on Co3D [34], ACID [23], and Real-Estate10k [68]. When possible we evaluate the models provided by the original works. Most closely related to our work in architecture is Zero1-to-3 [24] with the key difference being our addition of motion masking for training on video.

## 6.2 Novel View Synthesis

We observe improved performance on DTU and Mip-NeRF 360 on the standard NVS metrics (Tables 1 and 2). Our improvement on DTU is relatively small which is to be expected as the dataset consists of simple objects, with black backgrounds. On Mip-NeRF 360, which consists of real-world scenes, we see significant improvement. In particular, the other methods struggle to generate reasonable images from views that differ significantly from the input view. In Figure 3 we compare qualitatively to other recent works. We observe that Zero1-to-3 cannot generate full scenes and struggles to generate real objects as expected due its training data. ZeroNVS generates more plausible views, but is still considerably worse for more complex scenes.

## 6.3 3D Reconstruction

We present 3D scenes reconstructed from ODIN generated images along a trajectory (Figure 4). Quantitative comparison can be found in Table 3. For Google Scanned Objects [13] our method is comparable to Zero-1-to-3 [24] and outperforms other methods. Comparable performance to Zero-1-to-3 is expected as it was designed for synthetic objects. We compare with ZeroNVS for scene reconstruction on a held-out set of 360-1M (Table 4 in Appendix). Other methods are not capable of generating scenes therefore we only benchmark against this method.

Table 3: 3D reconstruction results on Google Scanned Objects [13].

| Method | MCC [54] | SJC-I [51] | Point-E [30] | Zero-1-to-3 [24] | ODIN (Ours) |
|---|---|---|---|---|---|
| **Chamfer Distance** ↓ | 0.1230 | 0.2245 | 0.0804 | 0.0717 | **0.0697** |
| **IoU** ↑ | 0.2343 | 0.1332 | 0.2944 | 0.5052 | **0.5328** |

# 7 Limitations and Broader Impact

The framework and method we presented in this work are a promising step towards large-scale 3D models, however there are some limitations to our approach. From a modeling perspective, the motion mask allows us to filter portions of scenes which have dynamic elements, however ideally we would like to learn to model the dynamic elements as well. Some progress has been made on 4D NeRF models which can move the camera view in both time and space, however generalized 4D models are largely unexplored.

Our work may have positive societal impact in the creation of 3D assets for AR and VR or downstream applications such as robotic navigation. From a negative perspective our work could be used to create fake images or inappropriate scenes.

# 8 Conclusion

In this work, we propose a scalable approach to constructing real-world multi-view data and show the merits of our model, ODIN, trained on the largest multi-view dataset, 360-1M, to date. Enabled by the scale, diversity, and long-range correspondences in 360-1M, ODIN demonstrates capabilities beyond those of previous methods in generating 3D-consistent novel views of real-world scenes with free camera movement. On novel view synthesis and 3D reconstruction benchmarks ODIN outperforms existing methods without fine-tuning to the target data. While ODIN shows impressive results, we believe that there is further potential in the use of 360-1M and 360° video for novel view synthesis as well as other domains such as video generation. For novel view synthesis an exciting next step would be to model dynamics to generate 4D scenes. We will open-source our code, models, and dataset.

## Acknowledgments and Disclosure of Funding

We would like to thank Kuo-Hao Zeng for his feedback on the manuscript. We acknowledge funding from NSF IIS 1652052, IIS 1703166, DARPA N66001-19-2-4031, DARPA W911NF-15-1-0543 and gifts from Allen Institute for Artificial Intelligence, Google and Apple. Sham Kakade acknowledges

funding from the Office of Naval Research under award N00014-22-1-2377. This work has been made possible in part by a gift from the Chan Zuckerberg Initiative Foundation to establish the Kempner Institute for the Study of Natural and Artificial Intelligence.

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

## A  Dataset Statistics

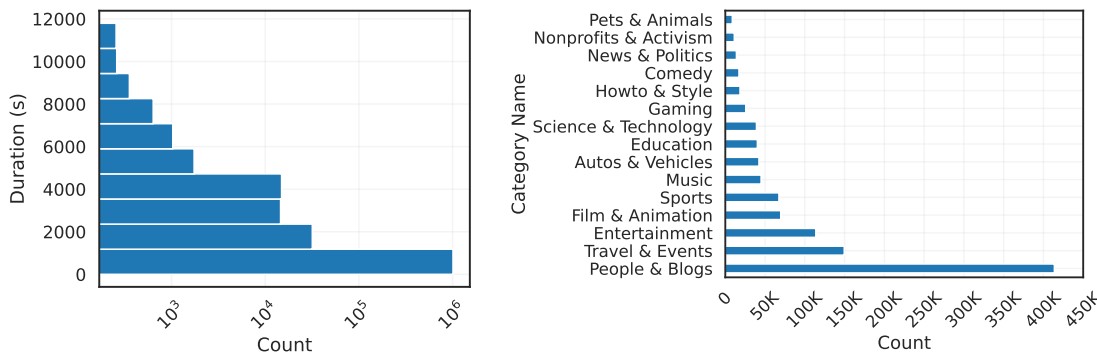

Figure 5: Video duration distribution in 360-1M.    Figure 6: Video categories' distribution in 360-1M.

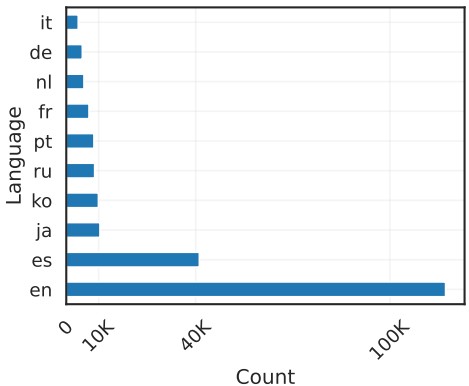

Figure 7: Video language distribution in 360-1M.

## B  Correspondence Examples

## C  3D Reconstruction Evaluation

Table 4: 3D reconstruction results on 360-1M [13]. Comparison with Zero 1-to-3.

| Method | Zero 1-to-3 | Our Method |
|---|---|---|
| **Chamfer Distance** ↓ | 0.1059 | **0.07992** |
| **IoU** ↑ | 0.3178 | **0.5267** |

## D  Safeguards for Data Release

Upon public release of our data we will require applications to obtain the links to the videos and meta-data.

## E  License for Data Release

YouTube videos are under fair use for research purposes and we provide only links to videos in the dataset.

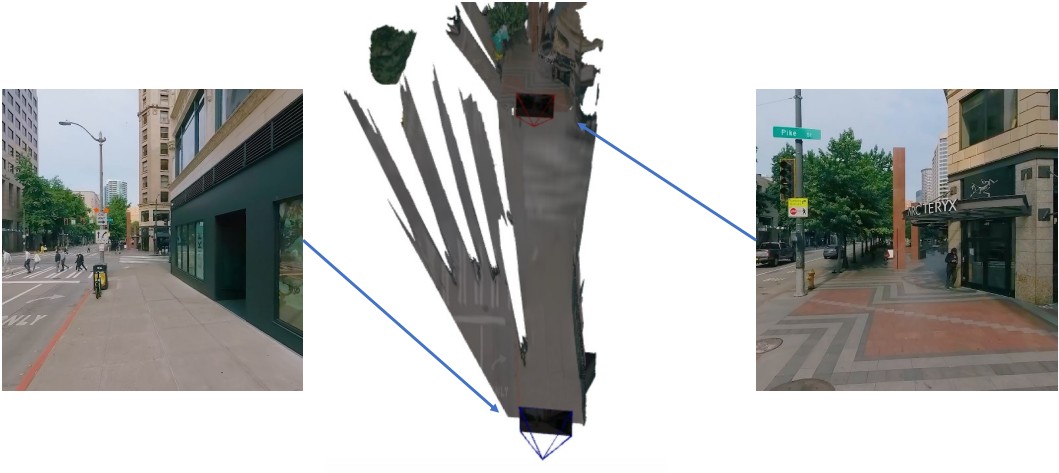

Figure 8: Example of long-range correspondence found automatically within 360-1M.

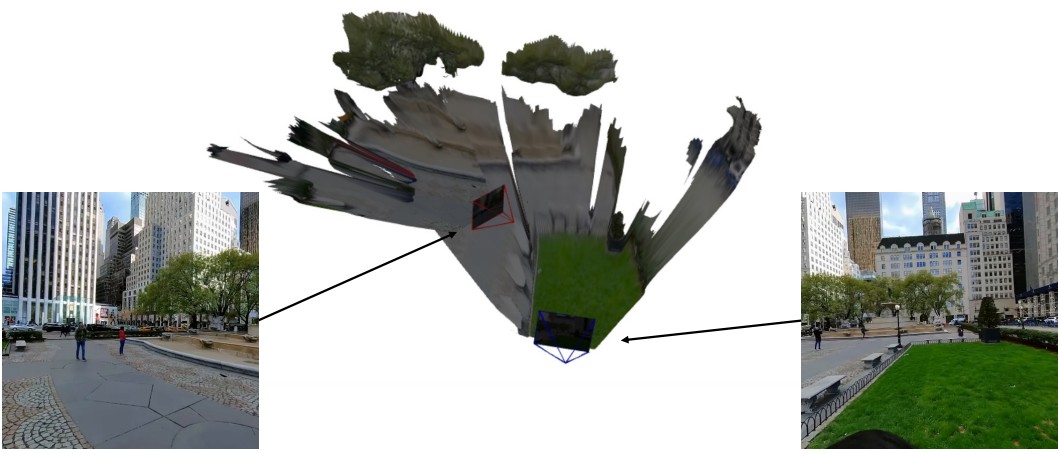

Figure 9: Example of long-range correspondence found within 360-1M.

## F   Training Details

We train ODIN, for 2 weeks on 16 A40's for 100 epochs. We used a batch size of 1024 where one sample consists of a frame correspondence. We use a learning rate of $1e-4$ with a constant learning rate. In general, if not otherwise specified we use the default hyper-parameters from [24]

## G   Ablations

In table 6 we show performance for various values of $/lambda$, the coefficient for motion masking detailed in section 5.2. In table 5 we show ablation over sampling various frames per second for 10k videos. We find that performance increases with higher FPS, and chose a reasonable balance between performance and compute cost at 1 FPS when scaling to the larger 1 million datasets.

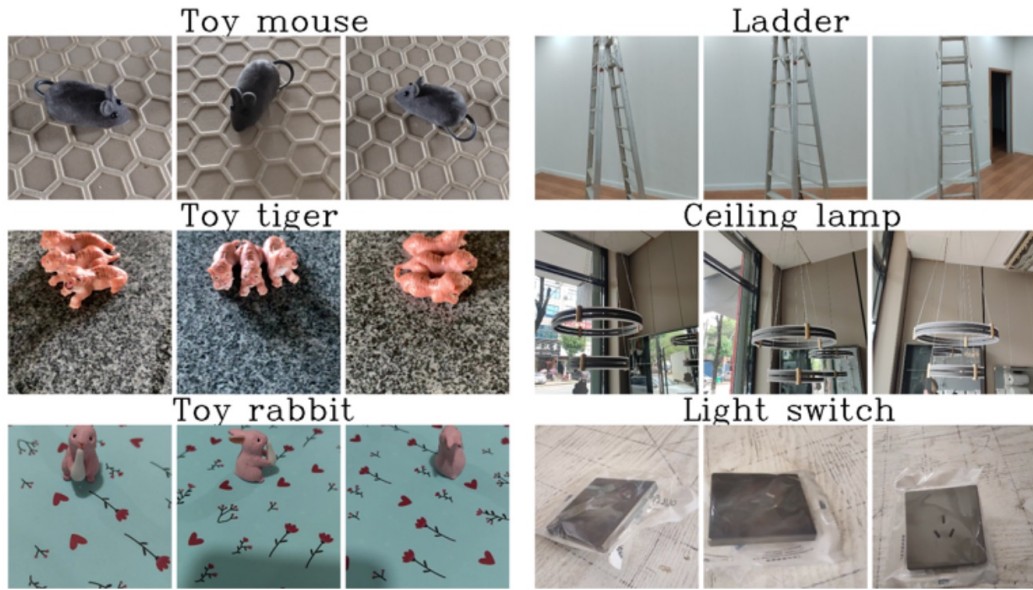

Figure 10: General example of correspondences from MVImageNet. Previously the largest multi-view dataset.

Table 5: Evaluation of LPIPS, PSNR, and SSIM at different frame rates (FPS) of sampling.

| FPS | LPIPS | PSNR | SSIM |
|---|---|---|---|
| 0.5 FPS | 0.488 | 15.88 | 0.492 |
| 1 FPS | 0.467 | 16.67 | 0.525 |
| 5 FPS | 0.461 | 16.85 | 0.539 |
| 10 FPS | 0.475 | 16.71 | 0.536 |

Table 6: Ablation study over $\lambda$ values for motion masking with novel view synthesis metrics.

| $\lambda$ | LPIPS | PSNR | SSIM |
|---|---|---|---|
| 0.1 | 0.498 | 12.31 | 0.366 |
| 0.5 | 0.467 | 14.73 | 0.402 |
| 1 | 0.378 | 16.67 | 0.525 |
| 2 | 0.395 | 14.94 | 0.431 |

