# OpenReview forum: "From an Image to a Scene: Learning to Imagine the World from a Million 360° Videos"
_NeurIPS.cc/2024/Conference — NeurIPS 2024 poster_

### Official Review · Reviewer_whPn · 2024-07-06

**Soundness:** 3
**Presentation:** 3
**Contribution:** 3
**Rating:** 7
**Confidence:** 4

**Summary:**

The paper introduces a large-scale, real-world, multiview, 360-degree outward dataset designed for static novel view synthesis and 3D reconstruction. To capture these data, the study develops a pipeline capable of identifying corresponding multiview frames from 360-degree outward videos with a fixed camera trajectory. To demonstrate the dataset's contribution and impact, the work proposes a diffusion-based model that achieves state-of-the-art performance using this dataset.

**Strengths:**

Strength
- The work introduces a real-world, multiview, 360-degree outward dataset that is significantly larger and more diverse than existing datasets for static novel view synthesis. This dataset is crucial for modern data-driven methods.

- The work presents a novel and efficient pipeline that utilizes Dust3R to find corresponding frames and graphs to maintain long-range correspondence for data collection. This innovative pipeline makes the data collection process scalable.

- The work proposes a diffusion-based model, ODIN, that leverages the video dataset for static novel view synthesis by incorporating motion masking and viewpoint conditions to achieve state-of-the-art performance. Additionally, the model can generate long-range novel views, further validating the dataset's positive impact.

**Weaknesses:**

Dateset
- The dataset is collected from 360 YouTube videos, captured beforehand by the camera operator. The camera trajectory (x, y, z) is controlled by the camera operator, thereby limiting the flexibility to select diverse viewpoints from different positions.

- The quality of the data is unknown and can be unstable across different data points because the author cannot investigate each video one by one or frame by frame. This represents a tradeoff between capturing one's own data and crawling data from the Internet. For instance, the author acknowledges in lines 178-179 that it is infeasible to guarantee the uniqueness of video content.

- Although the work claims that 1 fps is sufficient in lines 126-128, this claim lacks experimental support. The author should consider trying different fps rates and show the performance gap and computational time between them.

- Larger dataset may need more training time and computational resource.

Method
- One of the most challenging aspects that prevent people from using video for static scene tasks is the impact of motion over time (multiview inconsistency). The work proposes a motion mask method to mitigate this impact. However, this aspect lacks analysis and discussion. For instance, the importance of the hyperparameter in Eq. 3 is unknown.

- In Fig. 3, in the second row, the method is shown to synthesize humans as well. However, I wonder whether the motion masking has a negative impact on synthesizing dynamic objects when they are static.

Writing
- In lines 5-6, the claim may be too arbitrary. As mentioned in lines 87-90, there are some large-scale real-world datasets, such as CO3D, albeit with limited diversity. A related work, OmniObject3D, should also be mentioned.

- In lines 8-12, the phrase "...diverse views" may be too broad and should be softened. The proposed dataset is collected from pre-captured 360 YouTube videos. The camera trajectory is decided by the camera operator beforehand, and only the camera orientation can be controlled after that. Hence, compared to general multiview datasets, which can have multiple viewpoints from different locations at the same timestamp, the views of the proposed dataset are limited.

- Eq. 2 does not match the equation in Sec. 5.1. The equation in Sec. 5.1 contains an undefined term.

- I hope the author can define their 360 multiview videos at the beginning since there are several kinds of 360 videos. For instance, DiVa360 captures 360-degree inward videos with multiple cameras at different locations at the same timestamp.

**Questions:**

- I hope the author can verify that the viewpoints are diverse enough by discussing the difference between a multiview dataset that can capture images from different locations at the same timestamp and a dataset that can only change the camera orientation at the same timestamp.

- The dataset contains a large amount of data, which may vary in quality. I hope the author can discuss the impact of the low-quality data in the dataset. Is it negligible?

- Iines 178-179 mentions that it is infeasible to guarantee the uniqueness of video content. Hence, how do you make sure there is no overlap between training and testing set? How do you make sure the training dataset does not contain the testing cases in other dataset? Will it be unfair for the benchmark?

- The dataset contains diverse data. How is the data distribution in terms of motion? Will high-motion data, such as scenes with rain, have a negative impact?

- Lines 126-128 need experimental support. Please consider trying it on a subset of the dataset.

- The work uses Depth Anything as the depth estimator. This is a monocular depth estimator with two types of checkpoints pretrained on two different datasets. I wonder if this causes a domain gap in the proposed dataset when using different checkpoints. Do you finetune the estimator on your dataset? Additionally, it is unknown how good the multiview consistency is on your data with this estimator.

- What is the tradeoff between using this dataset and others? From the experiment section, we know that the dataset can increase performance. But how about the training time compared to existing datasets? Maybe the author can consider adding the training time from the appendix to the table.

- How sensitive and stable is the motion masking hyperparameter in Eq. 3? How should it be tuned?

- What is the performance when the input image contains many dynamic objects?

**Limitations:**

I agree with the limitation section in the paper.
- One potential limitation of the large-scale dataset is to maintain the data quality and high fps.
- Another potential limitation is to make sure there is no overlap between training and testing set.

---

> ### Author Rebuttal · Authors · 2024-08-07
>
> We thank the reviewer for their thorough review. We are happy that they find our dataset crucial for modern data-driven methods and find our correspondence search pipeline to be innovative. We have addressed comments and questions below and are happy to engage in further discussion. Note that we could not address all points within the character limit and will address the remaining questions during the discussion phase in the official comments.
>
> **The camera trajectory (x, y, z) is controlled by the camera operator, thereby limiting the flexibility to select diverse viewpoints from different positions.**
>
> We found that the increased scale and diversity of camera views in the 360-1M dataset compensates for this limitation as shown qualitatively and quantitatively in our results. The 1 million in-the-wild videos provides camera movements with extremely diverse trajectories, for example people sky-diving, climbing mountains, and walking many blocks through cities. Though each video has a fixed (x,y,z), the diversity of viewpoints across the data gives our model flexibility in generating from different viewpoints compared to models trained on previous datasets such as ObjaVerse and Co3D.
>
> **The quality of the data is unknown and can be unstable across different data points. For instance, the author acknowledges that it is infeasible to guarantee the uniqueness of video content.**
>
> While it’s true that we do not crawl every frame, we contend that the quality of the model is a strong indicator of the data quality. In our case we show that the scale and the quality of the 360-1M leads to improved model performance compared to previous datasets.
>
> **Although the work claims that 1 fps is sufficient in lines 126-128, this claim lacks experimental support.**
>
> We initially ran small-scale proxy experiments at various FPS before scaling to the full dataset. Performing such experiments at 1 million scale was computationally infeasible. We found that below 1 FPS there was no performance loss. We’ve also included the computational cost difference. Below we’ve included the results of our experiments on DTU and have added this to the appendix. For this experiment we kept the window size for correspondence searching fixed and used 50k videos from 360-1M.
>
> |          | LPIPS | PSNR  | SSIM |
> |----------|-------|-------|------|
> | 0.5 FPS  | 0.488 | 15.88 | 0.492|
> | 1 FPS    | 0.467 | 16.67 | 0.525|
> | 5 FPS    | 0.461 | 16.85 | 0.539|
> | 10 FPS   | 0.475 | 16.71 | 0.536|
>
> **Larger dataset may need more training time and computational resource.**
>
> We agree that considering computational cost is important. The task of finding corresponding pairs in video and estimating their relative pose was the main computational cost for us. By providing the image pairs and relative pose, other researchers can benefit from large-scale multi-view datasets without the computational burden of parsing a million videos.
>
> **One of the most challenging aspects is the impact of motion over time. This aspect lacks analysis. For instance, the importance of the hyperparameter in Eq. 3 is unknown.**
>
> We’ve included an ablation below over the hyperparameter lambda in Eq. 3 and have added discussion about choosing lambda to the appendix. We found that if the lambda value was too small, then the model would mask all objects in the scene and the reconstruction quality was poor. If the lambda value was too large, then smearing would occur near dynamic objects as the model was forced to predict the motion of the object.
>
> |                    | LPIPS | PSNR  | SSIM  |
> |--------------------|-------|-------|-------|
> | 0.1 = $\lambda$      | 0.498 | 12.31 | 0.366 |
> | 0.5 = $\lambda$     | 0.467 | 14.73 | 0.402 |
> | 1 = $\lambda$        | 0.378 | 16.67 | 0.525 |
> | 2 = $\lambda$        | 0.395 | 14.94 | 0.431 |
>
> **I wonder whether the motion masking has a negative impact on synthesizing dynamic objects when they are static.**
>
> Qualitatively we found that motion masking helped in reconstructing dynamic objects even when they are static such as parked cars. We hypothesize that due to the scale of the dataset, objects which can be both dynamic and static are seen multiple times in both settings.
>
> **As mentioned, there are some large-scale real-world datasets, such as CO3D, albeit with limited diversity. A related work, OmniObject3D, should be mentioned.**
>
> Thanks, we have added OmniObject3D to the related works. We’ve changed line 5-6 to be more concrete. Co3D is impressive and relatively large-scale, but orders of magnitude smaller than our proposed dataset (20 thousand vs 1 million videos).
>
> **In lines 8-12, the phrase "...diverse views" may be too broad.**
>
> Often datasets such as DTU, MVImageNet and Co3D have images taken from positions close in space around an object (1-5 meters apart). By diverse views we meant varying differences in camera pose (up to 50 meters apart and 360 degree rotation) and from various locations, not only around objects or landmarks. We will make it more precise what we mean by diverse viewpoints in the abstract and throughout the paper.
>
> **I hope the author can discuss the difference between a multiview dataset that can capture images from different locations at the same timestamp and one that can only change the camera orientation.**
>
> One limitation of a multi-view video dataset is that camera poses at different points in the (x,y,z) trajectory will have different timestamps. Learning novel view synthesis from dynamic scenes is still an open problem which temporal masking addresses to an extent, but there is still significant progress to be made. Current novel view synthesis (ZeroNVS, MegaScene) works seem capable of training on datasets where time and spatial location change simultaneously. Datasets which capture images from different locations at the same timestamp are ideal, but we believe this would be extremely difficult to manually collect at scale, and harnessing existing video data is an appealing alternative.

---

> > ### Author Response · Authors · 2024-08-07
> > **Continued Response to Reviewer whPn**
> >
> > Below we've included responses to the remaining questions and comments from reviewer whPn. We appreciate their detailed feedback and questions.
> >
> > **The dataset contains a large amount of data, which may vary in quality. I hope the author can discuss the impact of the low-quality data in the dataset. Is it negligible?**
> >
> > We found that low quality video frames such as blank frames, blurry frames, low resolution frames, or those with minimal camera movement were automatically filtered out in the Dust3r scoring phase. If the frames were low quality they naturally received a low score because the depth could not be well estimated. We verified this through manual inspection by randomly sampling frames which were selected by the Dust3r scoring mechanism.
> >
> > **Iines 178-179 mentions that it is infeasible to guarantee the uniqueness of video content. Hence, how do you make sure there is no overlap between training and testing set?**
> >
> > We ran deduplication over video titles, URLs, and thumbnails for all 1 million videos though this does not guarantee videos contain overlapping clips. The primary utility of a large multi-view dataset is for training models. We believe that datasets such as Mip-NeRF are better for evaluation.
> >
> > **Lines 126-128 [in theory we can align the views to look at the various regions of the scene to form multiple view correspondences] need experimental support. **
> >
> > Figure 8 and 9 in the appendix show qualitative paired frames from the dataset. Quantitatively we found 80,567,325 paired frames from the videos by aligning the views. Upon manual inspection we found that the frames were aligned to contain overlapping content. We will add more qualitative paired frames from the dataset to the appendix for readers view.
> >
> > **The work uses Depth Anything as the depth estimator. I wonder if this causes a domain gap in the proposed dataset.**
> >
> > We used the outdoor version of Depth Anything off-the-shelf. We found that the depth estimation did not need to be extremely precise, as small errors in the scale of the scene did not impact training the model.
> >
> > **What is the tradeoff between using this dataset and others?**
> >
> > We believe our dataset is complementary to other datasets. For example, ZeroNVS combines multiple datasets to train their model. We see no reason why our dataset could not be similarly combined with other multi-view datasets to train even better models. In terms of training, the training time scales proportionally with dataset size so more data means longer training. We will add a FLOPS comparison to the appendix.

---

> > > ### Comment · Reviewer_whPn · 2024-08-08
> > >
> > > I thank the authors for addressing all of my questions and concerns in the rebuttal. I also read the other reviews. I understand how challenging it is to propose a dataset. I personally consider the dataset pipeline to be part of the technical contribution. I am glad that the authors ran some experiments on a subset with ground truth to ensure the quality.
> > >
> > > If the authors are willing to share, I am curious about the plan to maintain this dataset. For dataset paper, I think long-term maintenance is much more important than the paper itself sometimes.
> > >
> > > After reading the rebuttal, I decide to change my score from weak accept to accept.

---

> > > > ### Author Response · Authors · 2024-08-08
> > > > **Thanks For the Discussion and Score Increase**
> > > >
> > > > We appreciate your diligence as a reviewer and the insightful feedback for improving our work! Our current plan is to host the links to the videos, the relative camera poses, the video meta-data, and the pretrained models through Hugging Face. Additionally we plan to release the data processing pipeline through github along with the model training code. If you have advice for long-term maintenance of datasets we appreciate your input.
> > > >
> > > > Also, we thank you for the rating increase.

---

> > > > > ### Comment · Reviewer_whPn · 2024-08-08
> > > > >
> > > > > That sounds good. About the long-term maintenance, I don't have too much experience with "super large-scale" dataset. I am not sure whether it is easy to find a place to store such a huge dataset in long term. Also, when I am taking about the maintenance, I am also talking about the update. Just, for example, Waymo released 9+ versions of its dataset [1] from 2019 to 2024, each with additional data or labels. But, anyway, this is not part of my reviews. I believe you all can figure out a way for the dataset.
> > > > >
> > > > > [1] Scalability in perception for autonomous driving: Waymo open dataset, CVPR 2020

---

> ### Comment · Reviewer_whPn · 2024-08-14
> **Growth Rate (Open question)**
>
> I just realized that my discussion is not public to the authors. Anyway, I copy and paste it here...
>
> I have an open question regarding data maintenance. The dataset is large-scale because it includes existing data accumulated over time from YouTube, in addition to the data collected through your pipeline. My concern is whether the data growth rate might slow down in the future. Perhaps the authors could consider plotting the total amount of data over time or by year, so that readers can understand the data growth rate. This could also help the authors estimate the expected time needed to collect a new large amount of data and update the dataset to a second version.

---

### Official Review · Reviewer_pxup · 2024-07-08

**Soundness:** 3
**Presentation:** 4
**Contribution:** 3
**Rating:** 7
**Confidence:** 4

**Summary:**

This paper considers novel view synthesis from a single image. The main contribution is a dataset sourced from 1 million 360 videos from youtube, which is used to create about 380 million pairs of images with corresponding relative poses. With this dataset the authors train a diffusion model similar to zero-1-to-3 [24], but for general scenes instead of objects, and real-world scenes instead of synthetic ones. State-of-the-art results are shown for novel view synthesis from single images.

**Strengths:**

- The main contribution of the paper is that it introduces a dataset for novel view synthesis that is significantly larger (380 million image pairs extracted from 1 million 360-videos) than existing datasets for the task, and considers whole scenes rather than single objects as e.g. objaverse.
- The authors train a diffusion model (ODIN) for novel view synthesis that is more general than existing work (zero-1-to-3 and follow up works which only consider single objects), and compared to scene based methods (zeroNVS), it is more accurate and general, permitting a larger set of possible relative poses (R,t). The improvements are mainly due to training on the larger dataset since the methodology, namely the diffusion model architecture and training, is very similar to zero-1-to-3.
- The proposed diffusion model, ODIN, along with the 3d reconstruction method dust3r can generate realistic-looking 3d reconstruction from a single image.

**Weaknesses:**

- There is a lack of qualitative examples. Specifically, a few things that would have been useful to show are 1) The generated images along the trajectories in fig. 4 in the form of both images and videos. Only the final 3d reconstruction by dust3r and not the intermediately generated images by ODIN are shown, 2) Images of the renderings of different methods in table 1 and 2, and 3) Generated images and the corresponding 3d reconstructions on Google Scanned Objects, so qualitative comparison corresponding to table 3 for all methods.
- There are a few missing references and as a result too strong claims in the paper. 1) “Long-Term Photometric Consistent Novel View Synthesis with Diffusion Models” (Yu et al. ICCV 2023), 2) “Geometry-Free View Synthesis: Transformers and no 3D Priors” (Rombach et al. ICCV 2021), and 3) “Look Outside the Room: Synthesizing A Consistent Long-Term 3D Scene Video from A Single Image” (Ren et al. CVPR 2022). These works, and also zeroNVS (which is cited) all address novel view synthesis from a single image, though trained on smaller datasets. ZeroNVS also constructs a 3d scene by training a NeRF with Score distillation sampling. Due to this I think e.g. the claim on line 48 that ODIN is the first to reasonably reconstruct a 3d scene from a single image is too strong.

**Questions:**

- I wonder how important the use of dust3r is for the 3d reconstruction. Is the confidence measure by dust3r used, and if so, how? I would imagine that it would be useful to get a consistent 3d reconstruction even if the views generated by ODIN are not consistent. By consistent I mean both that the views might not adhere to a specific camera model well enough for traditional sfm method to work well, or that the different views contain contradicting contents. Is that why there are some holes in the reconstructions in fig. 4 (e.g. the middle part in the leftmost 3d model of the cathedral), because it looks from the positions of the cameras that they should not be empty? What would happen if colmap was used on the generated images, would it still result in a reasonable 3d model?
- Did you notice any failure cases related to just conditioning on the previously generated image for the 3d generation? The camera motions in fig. 4 are fairly simple with little rotations and objects do not disappear and reappear, but if there would be more rotations I would imagine that it needs to be more carefully handled, e.g. as in the paper “WonderJourney: Going from Anywhere to Everywhere” (CVPR 2024).
- For 3d reconstruction, many methods (e.g. zeroNVS) use a NeRF model and score distillation sampling instead of direct 3d reconstruction from multiple generated images like it is done in this paper. Were any experiments like that performed?

**Limitations:**

This is adequately addressed

---

> ### Author Rebuttal · Authors · 2024-08-07
>
> We thank the reviewer for their thorough review. We are glad they appreciated the diversity and scale of our proposed dataset and the new capabilities of our resulting model. We provide responses to their comments and questions below and are happy to engage in further discussion.
>
> **A few things that would have been useful to show are 1) The generated images along the trajectories in fig. 4 in the form of both images and videos. Only the final 3d reconstruction by dust3r and not the intermediately generated images by ODIN are shown, 2) Images of the renderings of different methods in table 1 and 2, and 3) Generated images and the corresponding 3d reconstructions on Google Scanned Objects, so qualitative comparison corresponding to table 3 for all methods.**
>
> Great suggestion, we’ve included generated videos in the one page pdf for better visualization and have added 3D construction examples and further image examples to the paper.
>
> **There are a few missing references and as a result too strong claims in the paper ZeroNVS also constructs a 3d scene by training a NeRF with Score distillation sampling. Due to this I think e.g. the claim on line 48 that ODIN is the first to reasonably reconstruct a 3d scene from a single image is too strong.**
>
> Thanks for the suggestion. We’ve removed this sentence from the introduction and softened the language overall. Additionally we’ve added the missing references you’ve pointed us to.
>
> **I wonder how important the use of dust3r is for the 3d reconstruction. Is the confidence measure by dust3r used, and if so, how? I would imagine that it would be useful to get a consistent 3d reconstruction even if the views generated by ODIN are not consistent. By consistent I mean both that the views might not adhere to a specific camera model well enough for traditional sfm method to work well, or that the different views contain contradicting contents. Is that why there are some holes in the reconstructions in fig. 4 (e.g. the middle part in the leftmost 3d model of the cathedral), because it looks from the positions of the cameras that they should not be empty? What would happen if colmap was used on the generated images, would it still result in a reasonable 3d model?**
>
> For the 3D reconstruction portion colmap performs qualitatively the same. You’re correct that the hole in fig. 4 is due to low confidence from the Dust3R model in that region. In general, as long as enough images are generated,  experimentally, we found that colmap can perform similarly. To summarize, Dust3r paired with the graphical search is necessary for finding corresponding frames while constructing the dataset, but at inference the model is agnostic to the method for 3D reconstruction.
>
> **Did you notice any failure cases related to just conditioning on the previously generated image for the 3d generation? The camera motions in fig. 4 are fairly simple with little rotations and objects do not disappear and reappear, but if there would be more rotations I would imagine that it needs to be more carefully handled, e.g. as in the paper “WonderJourney: Going from Anywhere to Everywhere” (CVPR 2024).**
>
> Yes, as the reviewer mentioned the most common failure case is if an object comes in and out of occlusion it can be inconsistently generated by the model. With long-range camera trajectories more parts of the scene will come in and out of view which can lead to inconsistent generations, a typical failure mode of conditional NVS models. We found that the SDS anchoring introduced by ZeroNVS was effective in improving this consistency.
>
> **For 3d reconstruction, many methods (e.g. zeroNVS) use a NeRF model and score distillation sampling instead of direct 3d reconstruction from multiple generated images like it is done in this paper. Were any experiments like that performed?**
>
> In general the dataset and model are agnostic to the downstream 3D reconstruction method.
> We originally chose direct 3D reconstruction so that the geometry of the scene could be inferred in real time and we found it better for large-scale scenes and inferring geometry compared to distilling into a NeRF model.

---

> > ### Comment · Reviewer_pxup · 2024-08-12
> >
> > I thank the authors for providing these answers. I keep my accept rating.

---

### Official Review · Reviewer_Jg5T · 2024-07-12

**Soundness:** 3
**Presentation:** 3
**Contribution:** 3
**Rating:** 6
**Confidence:** 4

**Summary:**

This paper introduces a new approach for efficiently finding corresponding frames from diverse viewpoints from YouTube videos at scale. The resulting 360-1M dataset contains over eighty million frames from approximately one million videos. Additionally, the paper also builds on the 360-1M dataset by proposing the ODIN model, which synthesizes novel views and reconstructs 3D scenes from a single input image. The ODIN model is trained by leveraging the diverse viewpoints in the dataset to handle significant camera view changes and complex real-world scenes. The authors demonstrate the utility of the 360-1M dataset as well as the benefits of their proposed ODIN model by comparing it to state-of-the-art 3D reconstruction approaches, where it outperforms the latter by a significant margin.

**Strengths:**

1) The model figures are informative and especially helpful in helping the reader to understand the different stages of the data curation process as well as the intuition behind each stage. The paper is also well-organized and well-written.

2) The introduced algorithm to transform 360 degrees videos into the required multi-view format is especially significant. It opens up the possibility of scaling up available 3D object and scene datasets, which are often plagued by a lack of large-scale data. Furthermore, it is especially helpful that this algorithm appears to generalize well to videos from diverse subject categories.

3) The ODIN model also introduces a novel modeling objective which is conditioned on both rotation and translation. This differs from prior work which is unable to do so. This may be beneficial for generating more realistic samples of real world objects and scenes for further training.

**Weaknesses:**

1) The paper relies heavily on the 360-degree videos collected from YouTube, which may contain varying video qualities and resolutions. The described preprocessing steps does not fully account for this diversity in video quality, which may affect the quality of the extracted multi-view data negatively.

2) It is mentioned in line 220 that optimizing Equation 2 directly results in a degenerate solution and Equation 3 is introduced to mitigate this problem. However, there is no ablation to demonstrate this. It may help make the paper more comprehensive.

3) In Tables 2 and 3,  it is shown that ODIN outperforms other approaches on the LPIPS and PSNR metrics but not the SSIM metric consistently. However, there is no discussion of possible reasons for this discrepancy.

**Questions:**

Please look at the above-mentioned limitations.

**Limitations:**

Yes.

---

> ### Author Rebuttal · Authors · 2024-08-07
>
> We thank the reviewer for their thorough review. We are glad they found our algorithm for transforming 360° video into multi-view data to be especially significant and see its potential for scaling 3D datasets. We provide responses to their comments and questions below and would be happy to engage in further discussion.
>
> **The paper relies heavily on the 360-degree videos collected from YouTube, which may contain varying video qualities and resolutions. The described preprocessing steps does not fully account for this diversity in video quality, which may affect the quality of the extracted multi-view data negatively.**
>
> We experimented with filtering techniques such as CLIP filtering, where we fine-tuned a CLIP model to classify low quality frames from high-quality frames. For the fine-tuning data we hand-labeled 10k example frames. We found that the confidence scoring used to find image-pairs was sufficient for removing videos that were low in quality such as blank frames, corrupted or blurry frames, and static videos. Intuitively, frames that are low in quality are unlikely to be paired since relative pose and depth are difficult to estimate. For resolution we filtered out videos below a resolution of 2K. We also provide the available resolutions, view count, etc. for each video in the meta-data so users can filter videos based on various criteria.
>
> **It is mentioned in line 220 that optimizing Equation 2 directly results in a degenerate solution and Equation 3 is introduced to mitigate this problem. However, there is no ablation to demonstrate this. It may help make the paper more comprehensive.**
>
> Thanks for the suggestion. We’ve included an ablation below over the hyperparameter lambda in Eq. 3 and have added discussion about choosing lambda to the appendix. We found that if the lambda value was too small, then the model would mask all objects in the scene and the reconstruction quality was poor. If the lambda value was too large, then smearing would occur near dynamic objects as the model was forced to predict the motion of the object.
>
> |                    | LPIPS | PSNR  | SSIM  |
> |--------------------|-------|-------|-------|
> | $\lambda$ = .1      | 0.498 | 12.31 | 0.366 |
> | $\lambda$ = .5      | 0.467 | 14.73 | 0.402 |
> | $\lambda$ = 1       | 0.378 | 16.67 | 0.525 |
> | $\lambda$  = 2     | 0.395 | 14.94 | 0.431 |
>
> **In Tables 2 and 3, it is shown that ODIN outperforms other approaches on the LPIPS and PSNR metrics but not the SSIM metric consistently. However, there is no discussion of possible reasons for this discrepancy.**
>
> On line 247 we briefly note that previous works [1,2,3] have shown that SSIM and PSNR are not accurately correlated with better novel view synthesis. For example, in [3] figure 7 they show that a frame with uniformly grey pixels can outperform far more reasonable generations. In general, PSNR is sensitive to the low level pixel statistics which is tangential to the content of the image. We will better clarify this point in section 6.1.
>
> 1. Zero-Shot 360-Degree View Synthesis from a Single Image
> 2. Generative novel view synthesis with 3d-aware diffusion models
> 3. Nerdi: Single-view nerf synthesis with language-guided diffusion as general image priors

---

> ### Comment · Reviewer_Jg5T · 2024-08-13
>
> Thank you very much for your comprehensive efforts in addressing my concerns. In particular, I find your response on how and why the long-range image pairs are used particularly helpful. I also appreciate your efforts on the additional ablation experiments, given the time and computational constraints on your end as well as empirically quantifying the effects of using 1 FPS in other responses. After reading all of the reviewers' feedback as well as the corresponding responses from the authors, I will retain my original rating.

---

### Official Review · Reviewer_8VWi · 2024-07-13

**Soundness:** 3
**Presentation:** 3
**Contribution:** 3
**Rating:** 6
**Confidence:** 5

**Summary:**

This paper addresses the challenge of training models with 3D object understanding using large-scale real data, proposing the use of 360-degree videos as a scalable and diverse data source. Its main contributions include a 360-1M video dataset, an efficient method to convert 360-degree videos into multi-view data, and a novel diffusion-based model for view synthesis. The proposed model is evaluated on NVS and 3D reconstruction tasks compared to prior approaches.

**Strengths:**

1. The paper propose a novel large-scale real-world dataset collected from Internet along with an efficient pipeline to extract valid correspondence between frames. This dataset should be useful for future researches in the community.
2. The idea of utilizing 360 degree video to provide more correspondences is interesting.
3. Experiments demonstrate the proposed model is able to predict novel view images along a long trajectory, which is an important capability for large-scale scene/object understanding.
4. The paper is overall well written and easy to follow.

**Weaknesses:**

1. Besides its major contribution as a large-scale and open-world training dataset compared to ZeroNVS, the model primarily aligns with Zero-1-to3, which somewhat limits its technical contributions.

2.  Experiments:
[-] Could you provide a reason why Zero-1-to-3 is not compared on the DTU benchmark, which is suitable for Zero-1-to-3 given its focus on single objects placed on tabletops?
[-] I wonder how Zero-1-to-3 is applied in Fig. 3, as its camera pose involves both elevation and azimuth angles, which may differ from the video’s camera poses. Could the authors clarify how they calculate the camera pose?
[-] This paper presents quantitative results only for experiments on DTU and MipNeRF360 (Tables 1 and 2).  Including visual comparisons could help understanding.
[-] Why does Table 4 in the supplementary materials show a higher Chamfer Distance compared to ZeroNVS?

2. Others
[-] Considering the training requirements outlined in the appendix, it’s evident that training is highly resource-intensive, which is reasonable given the dataset’s scale. I wonder if there are potential methods to expedite this process, such as initializing from a pretrained Zero-1-to-3 model?
[-] Could the authors elaborate on how long-range image pairs contribute to the model? It’s challenging to distinguish correspondence in Fig. 9 from the appendix, as the images appear not to overlap. While such pairs may assist the model in extrapolating unseen regions, could they potentially hinder learning the viewpoint changes of observable content?
[-] The visualization in Fig. 3 could be enhanced by including relative changes in viewing angles, as it’s currently difficult for humans to find correspondence between the output and input view images.

**Questions:**

See above. Overall, I believe the proposed dataset and the demonstrated performance of the model hold promise for future research. I look forward to the rebuttal addressing my concerns.

**Limitations:**

Yes.

---

> ### Author Rebuttal · Authors · 2024-08-07
>
> We thank the reviewer for their thorough review. We are glad they find our model demonstrates important capabilities for large-scale scene understanding, the idea of using 360° video is interesting, and our dataset will be useful for future researchers. We provide responses to their comments and questions below and are happy to discuss further.
>
> **Besides its major contribution as a large-scale and open-world training dataset compared to ZeroNVS, the model primarily aligns with Zero-1-to3, which somewhat limits its technical contributions.**
>
> We contend that the technical contribution of this work goes significantly beyond a new dataset. We've enumerated our contributions below.
>
> -   We present a new technique capable of finding image-pairs from in-the-wild 360° videos. Corresponding image pairs are crucial for training novel view synthesis (NVS) models and current approaches such as colmap and hloc are not capable of extracting such pairs from large-scale video.
> -   We introduce temporal masking, a method which enables training on dynamic scenes which has been a limitation of novel view synthesis works.
> - We provide a new dataset consisting of over 1 million 360° videos, multi-view image pairs from the videos with their relative camera pose, and meta data for each video.
> -   Our model demonstrates novel capabilities in generating full, real-world scenes along long-range trajectories consisting of both rotation and translation compared to Zero1-to-3 which is limited to synthetic objects and camera rotations about the object.
>
> **Could you provide a reason why Zero-1-to-3 is not compared on the DTU benchmark, which is suitable for Zero-1-to-3 given its focus on single objects placed on tabletops?**
>
> We originally omitted Zero-1-to-3 since it is very similar to ZeroNVS and strictly worse in performance. We have added it back to Table 1 for completeness.
>
> **I wonder how Zero-1-to-3 is applied in Fig. 3, as its camera pose involves both elevation and azimuth angles, which may differ from the video’s camera poses. Could the authors clarify how they calculate the camera pose?**
>
> It’s true that Zero1-to-3 is limited to specific relative camera poses. We fit the azimuth, elevation angles, and radius of rotation to minimize L2 distance between the Zero1-to-3 camera pose and the target pose. Due to this limitation of Zero-1-to-3, we primarily focused our comparisons on ZeroNVS as it is designed for unconstrained camera movement, real-world scenes, and has a similar architecture.
>
> **This paper presents quantitative results only for experiments on DTU and MipNeRF360 (Tables 1 and 2). Including visual comparisons could help understanding.**
>
> We’ve included videos of our generation for better visual understanding in our one page pdf. Additionally we will add qualitative comparison for MipNeRF360 and DTU to the main paper.
>
>
> **Why does Table 4 in the supplementary materials show a higher Chamfer Distance compared to ZeroNVS?**
>
> Thanks for catching the type in the appendix. That should be a comparison with Zero-1-to-3. We’ve corrected the error.
>
> **I wonder if there are potential methods to expedite this process, such as initializing from a pretrained Zero-1-to-3 model?**
>
> We tried initializing from Zero-1-to-3 and ZeroNVS. We found that it led to slightly faster convergence, about ~10% fewer training iterations. The performance at convergence was the same when starting from both pretrained models.
>
> **Could the authors elaborate on how long-range image pairs contribute to the model?**
>
> The long-range image pairs in the training data are crucial for generating long-range trajectories and modeling large-scale scenes with our model such as in figure 1l. Models are limited to types of camera movements in their training distribution. We observe this with zero1-to-3 being limited to only rotating the camera around objects.
>
> **It’s challenging to distinguish correspondence in Fig. 9 from the appendix, as the images appear not to overlap. While such pairs may assist the model in extrapolating unseen regions, could they potentially hinder learning the viewpoint changes of observable content?**
>
> The trees and the steps in the background are shared by both images and Fig. 9 shows that Dust3r is capable of accurately finding the relative camera poses. Empirically we did not find that extrapolating to unseen regions decreased performance of the model for observable content.
>
> **The visualization in Fig. 3 could be enhanced by including relative changes in viewing angles, as it’s currently difficult for humans to find correspondence between the output and input view images.**
>
> Thanks for the suggestion, we will add this to figure 3 to improve the visualization.

---

> > ### Comment · Reviewer_8VWi · 2024-08-08
> > **Post-rebuttal**
> >
> > I thank the authors' efforts in addressing my concerns, especially over the experiment details. After reading the opinions from other reviewers, I raise my rating to weak accept.

---

### Author Rebuttal · Authors · 2024-08-07

We thank the reviewers for their insightful reviews and feedback. We are glad that the reviewers found our dataset to be crucial for modern data-driven methods [whPn] and our proposed pipeline for constructing multi-view data from 360° video to be innovative [whPn], interesting [8VWi] and especially significant for scaling 3D datasets [Jg5T]. Additionally we appreciate that the reviewers found our model is important for large-scale scene understanding [8VWi], introduces a novel modeling objective [ Jg5T], and achieves state-of-the-art for novel view synthesis [whPn].


**In the one page pdf we’ve included generated videos from our model to better visualize the outputs of ODIN. We recommend that the pdf be downloaded and viewed with adobe acrobat for the best viewing experience.**


Below we’ve addressed individual questions and comments and are happy to engage in further discussion with reviewers.

---

### Decision · Program_Chairs · 2024-09-25

**Decision:**

Accept (poster)

**Comment:**

The paper proposes a large-scale dataset of internet images and a method to extract correspondences between frames. After the rebuttal, all reviewers unanimously recommend acceptance. After reading the paper, the reviews and the rebuttal, the AC agrees that the paper should be accepted. The dataset, model, and method will be highly valuable to the community.

During the rebuttal, the authors have promised several changes and additions to the paper (e.g. include Zero-1-to-3, $\lambda$ analysis, discussion in sec 6.1, changes in the text, limitations, etc.) The AC strongly suggests that all these changes will be included in the final version of the paper.